# Alternative Non-Homologous End-Joining: Error-Prone DNA Repair as Cancer’s Achilles’ Heel

**DOI:** 10.3390/cancers13061392

**Published:** 2021-03-19

**Authors:** Daniele Caracciolo, Caterina Riillo, Maria Teresa Di Martino, Pierosandro Tagliaferri, Pierfrancesco Tassone

**Affiliations:** Department of Experimental and Clinical Medicine, Magna Græcia University, Campus Salvatore Venuta, 88100 Catanzaro, Italy; dan.caracciolo@libero.it (D.C.); caterinariillocr@gmail.com (C.R.); teresadm@unicz.it (M.T.D.M.); tagliaferri@unicz.it (P.T.)

**Keywords:** error-prone DNA repair, alternative non-homologous end joining pathway (Alt-NHEJ), genomic instability, synthetic lethality, DNA damage, PARP1, LIG3, PolQ

## Abstract

**Simple Summary:**

Cancer onset and progression lead to a high rate of DNA damage, due to replicative and metabolic stress. To survive in this dangerous condition, cancer cells switch the DNA repair machinery from faithful systems to error-prone pathways, strongly increasing the mutational rate that, in turn, supports the disease progression and drug resistance. Although DNA repair de-regulation boosts genomic instability, it represents, at the same time, a critical cancer vulnerability that can be exploited for synthetic lethality-based therapeutic intervention. We here discuss the role of the error-prone DNA repair, named Alternative Non-Homologous End Joining (Alt-NHEJ), as inducer of genomic instability and as a potential therapeutic target. We portray different strategies to drug Alt-NHEJ and discuss future challenges for selecting patients who could benefit from Alt-NHEJ inhibition, with the aim of precision oncology.

**Abstract:**

Error-prone DNA repair pathways promote genomic instability which leads to the onset of cancer hallmarks by progressive genetic aberrations in tumor cells. The molecular mechanisms which foster this process remain mostly undefined, and breakthrough advancements are eagerly awaited. In this context, the alternative non-homologous end joining (Alt-NHEJ) pathway is considered a leading actor. Indeed, there is experimental evidence that up-regulation of major Alt-NHEJ components, such as LIG3, PolQ, and PARP1, occurs in different tumors, where they are often associated with disease progression and drug resistance. Moreover, the Alt-NHEJ addiction of cancer cells provides a promising target to be exploited by synthetic lethality approaches for the use of DNA damage response (DDR) inhibitors and even as a sensitizer to checkpoint-inhibitors immunotherapy by increasing the mutational load. In this review, we discuss recent findings highlighting the role of Alt-NHEJ as a promoter of genomic instability and, therefore, as new cancer’s Achilles’ heel to be therapeutically exploited in precision oncology.

## 1. Introduction

### 1.1. Role of Genomic Instability in Tumorigenesis

Cancer is a multi-step disease in which different features, conferring malignant phenotype, are progressively acquired. In this context, genomic instability is considered as a major cancer promoting mechanism because, by generating random mutations, it enables the acquisition of tumor hallmarks for surviving within the microenvironment by Darwinian selection [1]. Different types of genomic instability have been described [2]: (1) microsatellite instability (MSI), in which expansion or contraction of oligonucleotide repeats are observed in microsatellite regions, due to deficiency of mismatch repair (MMR); (2) chromosomal instability (CIN), the most common, characterized by structural or numeric chromosomal aberrations [3]; (3) nucleotide instability (NIN), defined by the presence of base substitutions, deletions, or insertions. These mutation prone phenotypes could derive from increased cell sensitivity to exogenous or endogenous mutagens and/or from loss of mechanisms, which normally act as genome guardians, such as p53, driving genetically damaged cells into senescence or apoptosis. In this context, DNA repair machinery defects play a crucial role in fostering genomic instability by promoting accumulation of genetic changes which lead to neoplastic transformation or tumor progression [4].

### 1.2. DNA Repair Deregulation Triggers Genomic Instability

Although the molecular basis of genomic instability remains mostly undefined, recent evidence indicates that perturbation of DNA repair machinery could indeed play a critical role, since it allows DNA damage overload and tolerates select cells with increased proliferation and expression of pro-survival genes. Human cells are continuously exposed to DNA damaging agents from exogenous sources, such as chemical compounds, UV light, or ionizing radiations (IR), as well as from endogenous stimuli like oxidative or replication stress and telomere erosion [5,6]. These mutagenic events could result in single-strand DNA breaks (SSBs), the most common, and/or into more dangerous DNA double-strand breaks (DSBs). To preserve the genome integrity, different DNA repair systems cooperate to prevent deleterious sequela of DNA damage on cell progeny. In particular, there are two main pathways which operate in the faithful repair of DNA DSBs: (1) the classical non-homologous end joining (C-NHEJ) [7,8], and (2) the homologous recombination (HR) [9,10]. C-NHEJ could operate during all phases of the cell cycle and does not require a DNA template to repair DNA damage. C-NHEJ consists of a rapid phase, which works to repair simple lesions not requiring any DNA ends processing, and a slower phase, which instead is involved in complex DSBs repair. In the first step of C-NHEJ, DNA lesion is recognized by the Ku70/Ku86 complex, which in turn recruits DNA dependent protein kinase (DNA PKcs) at the site of DNA damage. Then, upon phosphorylation by DNA-PK, endonuclease activity of Artemis is activated leading to 3′ and 5′ DNA overhangs processing. At this point, simple DNA lesions can be directly repaired by XLF/DNA ligase IV/XRCC4 complex, which is responsible for DNA ends ligation. In contrast, complex DSBs (such as those deriving from ionizing radiation) require preliminary DNA ends processing operated by different proteins, like polynucleotide kinase (PNK), the flap endonuclease-1 (FEN-1) and DNA polymerase µ and λ, which are responsible for only limited sequence Alterations at the junction. On the other hand, HR requires a homologous DNA template to start the repair and for this reason it mainly operates during S and G2 phases of the cell cycle. In the first reaction, DSBs are recognized by CtIP and BRCA1, which in turn activate the recruitment of other HR components at the site of DNA lesion. DNA end is then resected at 5′ by Mre11, Rad50, Nbs1 (MRN complex), generating long 3′ single-stranded DNA (ssDNA) overhang, which is coated and stabilized by the replication protein A (RPA) and Rad51 nucleoprotein filament. Finally, after homology search, 3′-ssDNA tails invade sister chromatid leading to D-loop formation, which is followed by DNA synthesis until the Holliday junctions become resolved resulting in DSBs repair [10].

Beyond these two major pathways, other DNA repair systems ensure fidelity repair in presence of specific DNA lesions. For instance, Fanconi Anemia (FA) pathway, which includes almost 22 different proteins, operates to remove a critical barrier for DNA replication and genetic transcription, the DNA interstrand crosslink (ICL). In particular, after ICL detection, ubiquitylated FANCD2-I recruits an endonuclease complex which cleaves the DNA strand contiguous to the ICL and generates a DSB, which is repaired by FA-pathway dependent HR repair [11].

In addition to C-NHEJ and HR, which play a critical role as genomic guardians, several error-prone DNA repair pathways could also take over in the presence of DNA damage and impairment of high fidelity repair, triggering the development of cancer genomic instability. We here focus on the role of one of these pathways, named alternative non-homologous end joining (Alt-NHEJ) [12], as promoter of genomic lesions and, at the same time, as cancer’s Achilles’ heel to be exploited with new synthetic lethality approaches in precision oncology.

## 2. Alt-NHEJ and De-Regulation in Cancer

### 2.1. Background on Alt-NHEJ Machinery Repair

The Alt-NHEJ repair, also known as microhomology-mediated end joining (MMEJ) pathway, requires from 2 to 20 nucleotides of sequence homology at DNA ends of DSBs to start repair [13,14]. In the first step, PARP1 recognizes DNA breaks [15] and activates DNA End Resection by MRN/CtiP complex [16], resulting in exposition of microhomology sequence at repair site [17]. Next, DNA ends are bridged and aligned via the short microhomologies, and non-homologous 3′ tails are digested by ERCC1\XPF nucleases. Resulting gaps within DNA strands are next filled by PolQ-mediated DNA synthesis [18], and then DSBs are finally joined by DNA Ligase III/XRCC1 complex [19,20]. In the absence of more effective DNA Ligase III, DNA Ligase I could also take over to catalyze the final step of DNA ends ligation [21] (Figure 1).

Recent experimental evidence underlines the role of Alt-NHEJ as back-up pathway when C-NHEJ or HR fail during DNA repair steps [13]: (a) DSB sensing, via PARP1/Ku competition for DNA ends [22]; (b) gap filling, through PolQ-mediated inhibition of HR by direct interaction with RAD51 [23] and removal of RPA from resected DSBs [24]; (c) DNA ligation, due to mutually exclusive activity of C-NHEJ dependent DNA ligase IV and Alt-NHEJ dependent DNA ligase III [21]. By contrast, functional HR or NHEJ directly suppresses error-prone Alt-NHEJ repair [25,26]. Furthermore, FA pathway deficiency could indirectly decrease Alt-NHEJ by increasing Ku dependent C-NHEJ [27]. Overall, these data show that multiple mechanisms work together to prevent dangerous consequences of aberrant activation of Alt-NHEJ on genomic stability.

However, beyond the role of backup pathway, a physiologic role for Alt-NHEJ is also demonstrated in DSB induced by ionizing radiation [28] or in mitochondrial DNA (mtDNA) metabolism, since DNA Ligase III is the principal DNA ligase of mitochondria [29].

Alt-NHEJ could generate erroneous repair, which in turn fosters genomic instability with different mechanisms. First, Alt-NHEJ does not operate with a DNA template strand as HR, and therefore it cannot restore original DNA sequence. Second, the gap filling is carried out by PolQ, which is responsible for erroneous nucleotide insertions due to low fidelity transferase activity and ssDNA microhomology-primed iterative synthesis. Indeed, PolQ is frequently insufficiently processive to complete repair of breaks in microhomology-poor regions. For this reason, aborted synthesis induces additional rounds of microhomology search, annealing, and synthesis, which generates sufficiently long *de novo* microhomologies to resolve broken ends [30,31]. Third, large deletions are generated by endonuclease/exonuclease complex to expose microhomologies sequence [32]; fourth, N-terminal zinc finger domain of DNA ligase III could catalyze the joining of unrelated DNA molecules, thus promoting translocations. In particular, this event is facilitated by high flexibility and distinct DNA binding domain features of DNA ligase III. Indeed, structural and mutational analyses indicate a dynamic switching between two nick-binding components of DNA ligase III, the ZnF-DBD and NTase-OBD, which could allow simultaneous binding of two different DNAs to stimulate intermolecular ligations (“jackknife model”) [33].

### 2.2. Transcriptional and Post-Transcriptional Alt-NHEJ Regulation

Experimental evidence indicates that Alt-NHEJ repair is finely regulated at transcriptional and post-transcriptional levels. In particular, different transcription factors exert their crucial role in tumorigenesis also by fostering Alt-NHEJ mediated genomic instability. For example, in BCR-ABL and FLT3 positive leukemias, c-MYC was demonstrated to induce the expression of LIG3 and PARP1 by increasing their transcription. This event led to increased Alt-NHEJ activity resulting in erroneous DNA repair characterized by high frequency of large deletions. Furthermore, c-MYC could promote Alt-NHEJ repair also by repressing the expression of LIG3 and PARP1 targeting microRNAs, such as miR-22, miR-27a, miR-34a, and miR-150. Consistently, c-MYC knock-down and/or c-MYC–regulated miRNAs overexpression was able to reduce ALT-NHEJ activity in FLT3/ITD- and BCR-ABL1-positive cells, thus indicating a master regulator role of c-MYC in genomic instability promotion [34], by Alt-NHEJ repair induction. More recently, an important role in Alt-NHEJ regulation was also demonstrated for long non-coding RNAs (LncRNAs). For example, in hepatocellular carcinoma (HCC) the lncRNA *lncPARP1*, which was significantly up-regulated in HCC patients, directly increased the expression of its target PARP1 acting *in cis*, thus triggering genomic instability and disease progression [35]. Similarly, another oncogenic lncRNA MALAT1 [36,37] seems to play an important role in Alt-NHEJ regulation. In particular, by using RNA antisense purification-mass spectrometry (RAP-MS) and ribonucleoprotein immunoprecipitation (RIP) strategy, a direct binding of MALAT1 to PARP1 and LIG3 was found. Importantly, MALAT1 inhibition by antisense oligonucleotide approach led to DNA damage and apoptosis, thus suggesting the existence of a novel level of complexity in the Alt-NHEJ regulation exerted by a lncRNA-protein complex, with a strong impact in proliferation and survival of cancer cells.

## 3. Alt-NHEJ Involvement in Tumors: An Overview

In the following sections we will describe the involvement of Alt-NHEJ in onset, progression, and acquisition of drug resistance of different tumors.

### 3.1. Ovarian Cancer

About 50% of high grade serous ovarian cancers (HGSOCs) display genetic or epigenetic alterations in the HR pathway. Similar alterations are less often associated to non-serous histology including clear cell, endometrioid, and carcinosarcomas [38]. Germline *BRCA1* and *BRCA2* mutations have been identified in 14–15% of all ovarian cancers while somatic *BRCA1* and *BRCA2* mutations are found in 6–7% of high grade serous EOCs [39]. FA/HR deficiency is an important therapeutic target in ovarian cancer, since it could be therapeutically exploited by the use of platinum agents [40] as well as by PARP inhibitors (PARPis) [41], thus confirming Alt-NHEJ addiction of this disease. Interestingly, a critical role of PolQ is been highlighted by recent studies showing that HR-deficient cells displayed higher levels of PolQ [23]. Consistently, PolQ knockdown or its pharmacological inhibition by Novobiocin induced synthetic lethality in these cells, further indicating Alt-NHEJ as promising target in HR deficient tumors.

### 3.2. Breast Cancer

HR deficiency occurs in up to 40% of familial and sporadic breast cancer [42]. *BRCA1/BRCA2* mutations account for the majority of hereditary breast cancers, representing about 5–7% of all unselected breast cancers. *BRCA1* mutations are often observed in TNBC tumors, while *BRCA2* mutations are mostly associated with ER-positive subgroup [43]. It has been also demonstrated that some sporadic breast cancers harbor defects in the HR and FA pathway, in the absence of a germline *BRCA1* or *BRCA2* mutation, a condition referred as BRCAness [44]. Indeed, beyond *BRCA1⁄2*, the main FA/HR affected genes are represented by *CHEK2*, *ATM*, *BRIP1*, *PALB2*, *PTEN*, *NBN*, *RAD51C*, *RAD51D*, *MSH6*, *PMS2*, and *FANCD2* [42]. Overall, current evidence indicates that, in the setting of *BRCA1*-defective tumors, HRD is an important therapeutic target in breast cancer, since it could be therapeutically exploited by the use of conventional drugs, including platinum agents [45,46,47] as well as PARPis, thus suggesting a relevant role of Alt-NHEJ in this disease. Indeed, an up-regulation of LIG3 and PARP1 was found in breast cancer cell lines as compared to normal breast epithelial cells. Furthermore, tamoxifen- and aromatase-resistant derivatives cells and hormone-receptor negative cells showed higher Alt-NHEJ activity, and reduced steady state levels of C-NHEJ proteins such as LIG4 and DNA-PK. In this context, combination of PARPis and DNA ligases inhibitor induced high cytotoxic activity, further indicating that Alt-NHEJ is a promising target in breast cancers resistant to standard therapies [48].

### 3.3. Neuroblastoma

Neuroblastoma cells exhibit high degree of chromosomic aberrations such as deletion and translocations, which predict poor survival and drug resistance [49]. Interestingly, recent experimental evidence suggests a pivotal role exerted by Alt-NHEJ in fostering genomic instability of neuroblastoma [50]. Consistently, up-regulation of Alt-NHEJ components DNA ligase III, DNA ligase I, and PARP1 was showed, as compared to C-NHEJ proteins DNA ligase IV and Artemis, which instead were downregulated. Furthermore, the authors demonstrated that *MYCN* overexpressing neuroblastoma cells are addicted to Alt-NHEJ repair for survival. Indeed, DNA ligase III, and DNA ligase I inhibition by L67 and PARP1 inhibitor treatment, led to DNA damage overload and finally neuroblastoma cell death. Furthermore, Alt-NHEJ was shown to be involved also in human neural crest stem cell (NCSC) neoplastic transformation by mediating *MYCN* pro-tumorigenic activity in neuroblastoma precursors [51].

### 3.4. Acute Leukemias

PARP1 and LIG3 are found up-regulated in acute myeloid leukemia (AML) patients as compared to healthy individuals, and most importantly, their expression was strictly associated with chromosomal translocations occurrence [52]. In particular, Alt-NHEJ repair is hyper-activated in FLT3/ITD-positive AML, resulting in high genomic instability [53]. Consistently in this cellular context, C-NHEJ proteins were downregulated while DNA ligase III was overexpressed in FLT3-positive AML. Importantly, FLT3 inhibitor led to Alt-NHEJ repair activity reduction, increased repair errors, and reduced genomic instability in FLT3-positive AML. Similarly, mutated KRAS T-cell acute lymphoblastic leukemia (T-ALL) cells showed up-regulation of LIG3 and PARP1 and, consequently, Alt-NHEJ repair hyper-activation [54]. Notably, targeting of Alt-NHEJ pathway by PARPis selectively sensitizes KRAS-mutant leukemic cells to cytarabine and daunorubicin, thus overcoming resistance to apoptosis mediated by oncogenic KRAS.

### 3.5. Chronic Myeloid Leukemia

In chronic myeloid leukemia cells, BCR-ABL fusion proteins have been shown to induce high production of ROS leading to DSBs generation, which undergoes repair by Alt-NHEJ. Indeed, steady state levels of core C-NHEJ components such as Artemis and DNA ligase IV were downregulated while DNA ligase III and WRN was upregulated in CML cells. Notably, siRNA-down regulation of DNA ligase III and WRN led to impairment of repair efficiency and to strong increase of DNA damage [55].

### 3.6. Multiple Myeloma

Multiple myeloma (MM) is strongly characterized by genomic instability, which leads to proliferation of malignant plasma cells with complex karyotypic alterations. In this context, our group demonstrated a pivotal role exerted by LIG3-mediated Alt-NHEJ in promoting genomic instability and survival of MM cells [56]. Consistently, higher LIG3 mRNA expression was found to correlate with poor overall survival and progression free survival in MM patients, and was associated to high-risk cytogenetic alterations, disease progression and relapse. Notably, LIG3 knockdown induced high DNA damage increase thus leading to MM cell death in vitro and tumor growth inhibition in vivo. To investigate the mechanism leading to up-regulation of LIG3 in MM, we focused on microRNAs, given their critical role exerted in MM pathogenesis [57,58,59,60,61]. Consistently, miR-22 enforced expression down-regulated LIG3 mRNA and protein levels and inhibited Alt-NHEJ repair. Importantly LIG3 reduction induced by miR-22 replacement increased DNA DSBs leading to MM apoptotic cell death and sensitization to Bortezomib. All together, these findings suggest that MM cells are addicted to Alt-NHEJ repair, thus unrevealing a novel mechanism of genome stability regulation and survival in MM. Consistently with its involvement in Alt-NHEJ repair, our group also provided pre-clinical evidence indicating that PARP1 is up-regulated in MM patients, where it associated to poor outcome, and notably, that PARP1 knockdown or its pharmacological inhibition by Olaparib led to anti-MM activity in vitro and in vivo [62]. Interestingly, in silico analysis suggested that high MYC expression could correlate with sensitivity to PARPis in MM. Therefore, we demonstrated that MYC promotes PARP1-mediated repair in MM and that anti-proliferative effects exerted by PARP inhibition are mainly observed in MYC-proficient MM cells, thus identifying a novel potential predictive biomarker of PARPis’ sensitivity in MM. Our findings are consistent with the evidence of HRD in MM samples as evaluated by next generation sequencing studies. Overall, these data provide the proof-of-concept for the study of PARPis in MYC-driven MM, particularly in the relapsed disease, as single agent as well as in combination with Bortezomib, exploiting the capability of this drug to induce HR down-regulation and therefore synthetic lethality when combined with PARPis [63].

## 4. Targeting Alt-NHEJ Repair

### 4.1. Drugging Alt-NHEJ Major Proteins

Deficit of DNA repair pathways, which normally operate as genome guardians, leads to increased genomic instability and in turn to tumorigenesis or disease progression. In cancer cells, HR or NHEJ deficiency is compensated by Alt-NHEJ hyperactivation, which allows the toleration of DNA damage overload produced by increased error rate. These findings suggest that targeting a backup repair pathway to which cancer cells are addicted could be exploited to selectively kill tumor cells while sparing normal cells, making therefore Alt-NHEJ a promising target for the treatment of DNA damage response (DDR)-deficient cancer cells [64].There are three main strategies to target Alt-NHEJ repair: (1) inhibiting the first step by PARPis, (2) preventing DNA synthesis at gaps targeting PolQ, and (3) blocking the final step of DNA end joining by DNA ligases inhibitors. For the first approach it must be considered that PARP family includes different enzymes catalyzing the synthesis of poly (ADP-ribose) polymers, which are then added onto target molecules in a process named PARylation. Among these, PARP1 is the most involved in Alt-NHEJ repair and, in particular, in the first step, where DNA damage is recognized. Indeed, after DNA binding, PARP1 catalytic function is activated and PARylation leads to the recruitment of DNA repair effectors [65], thus driving the start of Alt-NHEJ process. PARPis are designed as competitors of NAD+ for catalytic domain of PARP [66] (Table 1). Olaparib [67], Rucaparib [68], and Niraparib [69] have received clinical approval for the treatment and maintenance of ovarian cancer with germline BRCA mutation (gBRCAm). Olaparib is also approved for the treatment of gBRCAm breast cancers [69], a subset of gBRCAm pancreatic cancer patients [70], and patients with deleterious or suspected deleterious germline or somatic HR repair gene-mutated metastatic castration resistant prostate cancer (mCRPC) [71]. Talazoparib is approved for the treatment of deleterious germline BRCA-mutated HER2 negative metastatic breast cancer [72]. Finally, rucaparib also received breakthrough clinical designation by FDA for the treatment of adult patients with a deleterious *BRCA* mutation (germline and/or somatic)-associated mCRPC who have been treated with standard therapies [73]. The biological rationale that underlies the use of PARPis in BRCA-mutated cancers is not completely understood [74,75]. Indeed, different mechanisms have been hypothesized to explain the strong activity of PARPis in HR-deficient tumors [76]. The first model proposes PARPis as inhibitors of BER-dependent repair of SSBs, which are converted to DSBs unrepaired in cells’ carrier of homologous recombination deficiency (HRD). However, the absence of evidence of SSB occurrence after PARPi treatment, casts doubt on this model. In the second model, it has been hypothesized that upon PARPi treatment, PARP1 is trapped at sites of DNA damage, resulting in cytotoxic complex for HRD cells. However, this model also fails to explain why PARP1 knockdown also selectively induces cell death in BRCA1/2 mutated cells. Finally, the most interesting model suggests that PARPis exert their activity by blocking addiction of HRD cells to Alt-NHEJ repair, leading to increase of DNA DSBs and apoptotic cell death.

Regarding the second approach, it has to be considered that human cells express about 15 characterized DNA polymerases [77] which mainly operate in DNA replication and repair by catalyzing the synthesis, using a template, of complementary nucleotides to the 3′ end of DNA strands.

PolQ is the principal DNA polymerase of Alt-NHEJ, wherein it contributes to error-prone repair by high rate of abasic (AP) site insertion and robust trans-lesion synthetic activity. It is upregulated in several types of cancer where it predicts poor prognosis [78,79]. In particular, HRD cells are addicted to PolQ-mediated MMEJ, and consistently PolQ inhibition in these tumors leads to cell death [23]. In addition to C-terminal DNA polymerase domain, PolQ exhibits a N-terminal domain with a helicase-like domain having DNA-dependent ATPase activity. A large-scale small-molecule screen identified four potent PolQ ATPase inhibitors, such as mitoxantrone (MTX), suramin (SUR), novobiocin (NVB) and aurin-tricarboxylic acid (AUR). Among these, only NVB showed high specific inhibition of PolQ and importantly high anti-tumor pre-clinical activity in different HR-deficient cancer models (bioRxiv 2020.05.23.111658). 

For the third approach it needs to be underscored that there are at least three human DNA ligases [80] that are responsible for ATP-dependent DNA end-joining in the final step of DNA repair: (a) DNA ligase I, mainly involved in DNA replication [81], with minor roles in excision repair pathways (BER and NER), and Alt-NHEJ repair; (b) DNA ligase IV, which joins DSBs during c-NHEJ [82]; (c) DNA ligase III that plays a pivotal role in Alt-NHEJ repair (nuclear isoform) [83] and in mitochondrial DNA replication and repair (mitochondrial isoform) [84]. DNA ligases have a similar catalytic region consisting of oligonucleotide/oligosaccharide binding-fold (OB-fold), adenylation (AdD), and DNA-binding domains (DBD) [85]. In contrast to DNA ligase I and DNA ligase IV, DNA ligase III has a unique N-terminal zinc finger domain (ZnF) [33] which binds to DNA breaks enhancing the joining of DNA ends. Furthermore, it is demonstrated that ZnF domain plays a crucial role in intermolecular ligation of unrelated DNA molecules, thus promoting chromosome translocations [33]. Actually, the strategies of DNA Ligases inhibition are based on targeting their shared DBD, in the aim to prevent DNA ends recognition and finally block DSBs repair [86]. In particular, two different small molecules have shown to inhibit DNA Ligase III: L67, which blocks DNA ligase I and III, and L189 which instead inhibits DNA ligases I, III and IV. Notably, these compounds displayed either direct cytotoxic activity and/or sensitize different cancer cell lines to DNA damaging agents. However, further studies are needed to identify and characterize selective DNA Ligase III inhibitors, to efficiently inhibit error-prone Alt-NHEJ repair.

### 4.2. Alt-NHEJ Could Stimulate Immune Recognition of Tumor Cells

Errors generated during Alt-NHEJ repair could promote also immune recognition and immune destruction of cancer cells. Indeed, recent experimental evidence indicates that genomic instability triggers immunogenicity [87] of cancer cells by different mechanisms: (1) neo-antigen generation [88], (2) cGAS-STING pathway activation [89], and (3) immunological cell death induction [90]. However, these events are counterbalanced by the increased expression of immune-checkpoint CTLA-4 and PD-L1 to avoid detection and destruction by the immune system [91]. Indeed, immune check point inhibitors (ICIs) exerted strong anti-tumor effects in tumors characterized by high degree of genomic instability, such as melanomas and lung cancer [92,93]. As above mentioned, HR deficiency leads to activation of error-prone Alt-NHEJ repair which in turn fosters higher immunogenicity. Indeed, HR-defective ovarian and breast cancers display high mutations’ burden, neoantigens load and increased tumor infiltration by CD3+ and CD8+ lymphocytes [94]. Moreover, up-regulation of cGAS-STING activation markers is observed in HR-deficient as compared to HR-proficient cancer cells [95]. Mechanistically, it is hypothesized that deficiency of HR lead to up-regulation of error-prone MRE11 which activates Alt-NHEJ driven repair and innate immune activation by the STING pathway [96]. Overall, these findings propose hyper-activation of Alt-NHEJ not only as therapeutic target exploitable by the use of DDR inhibitors, but also as potential biomarker for high response to immune-oncology treatment of human cancer (Figure 2).

## 5. Conclusions and Perspectives

Tumor cells experience an overload of replication stress which leads to DNA damage [97]. In these circumstances, cancer cells often rely on error-prone DNA repair that allows survival by increasing the mutagenic rate, which in turn promotes genomic instability, disease progression and drug resistance [98,99]. In this review, we describe the pivotal role exerted by Alt-NHEJ, an error-prone DNA repair acting as back-up process when major DSB repair pathways fail. This is often observed in HR-deficient tumors which become highly addicted to Alt-NHEJ, thus providing the rationale for the design of novel therapeutic strategies to hit tumor cells with this DNA repair aberration, sparing normal cells from toxic side effects. One example is offered by the synthetic lethality occurring in *BRCA1* or *BRCA2* mutated cancer after PARPis treatment, the first therapeutic approach based on DDR inhibition [75]. More recently, the experimental data indicating that inhibition of PolQ is highly toxic in HR-deficient cells further confirmed the relevance of Alt-NHEJ addiction in HRD background [23]. However, the identification of predictive biomarkers of Alt-NHEJ inhibitors activity in cancer represents a challenge to exploit novel DDR vulnerabilities [100]. In this context, huge opportunities could derive from NGS technologies and pharmacogenomics, which allowed identifying mutational signatures and variant alleles [101,102,103,104] associated with high Alt-NHEJ repair activity. Furthermore, there is recent experimental evidence that cancers without HR deficiency, such as small cell lung cancer (SCLC) could also respond to PARPis. Potential biomarkers of such condition, termed “PARPness” [105], include: (1) loss of RB1 and TP53 expression combined with MYC overexpression, which leads to high replication stress and higher reliance on PARP1 repair for cell survival [106]; (2) high tumor PARP1 expression, resulting in lethal levels of PARP trapping; and (3) *IDH1* mutations which reduce production of NAD+ required for PARP1-mediated DNA repair [107]. Furthermore, it is becoming clear how error-prone repair pathways also represent potential biomarkers for high response to immunotherapy of human cancer. We can conclude that in the future error prone Alt-NHEJ DNA repair will play a major role as a target of selective therapeutics and will also provide novel decision making biomarker for immunotherapy approaches.

## Figures and Tables

**Figure 1 cancers-13-01392-f001:**
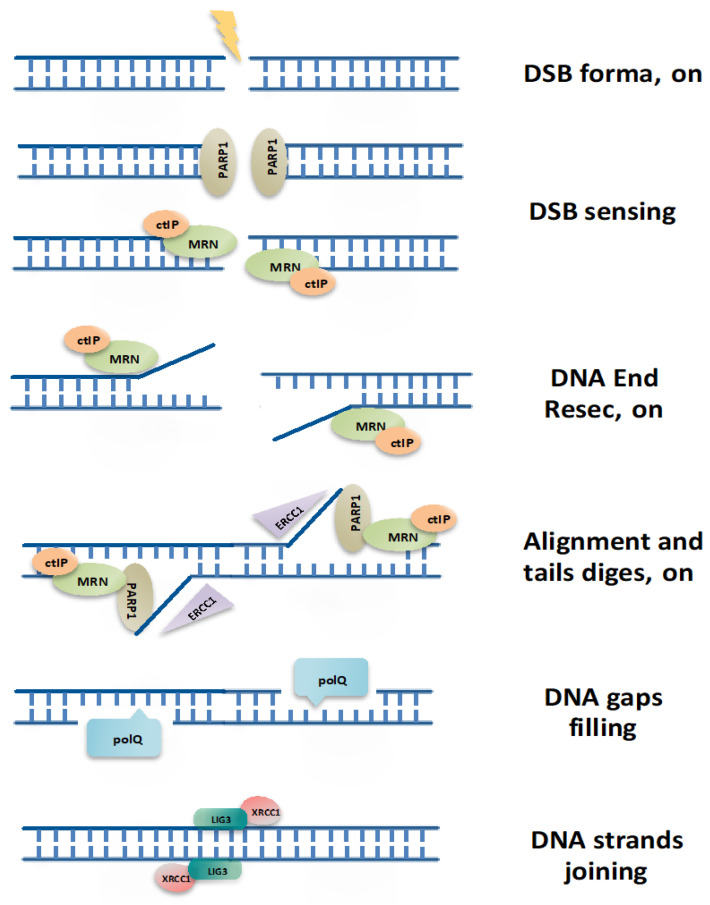
Alt-NHEJ machinery. Different steps and core components involved in DSB repair by Alt-NHEJ pathway. Alt-NHEJ, alternative non-homologous end joining; DSB, double-strand break.

**Figure 2 cancers-13-01392-f002:**
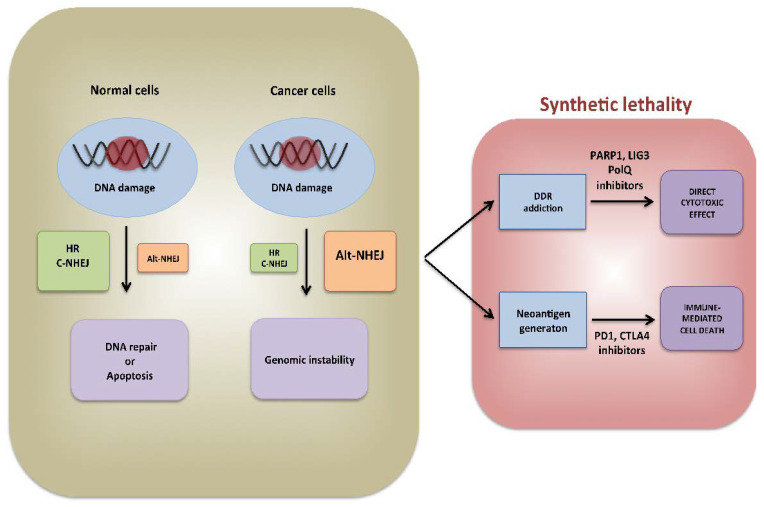
DNA repair dysregulation as cancer driver, therapeutic target, and biomarker of immunotherapy sensitivity. In normal cells, HR and c-NHEJ pathways work together to avoid dangerous effects of unrepaired DSBs. In cancer cells DSBs are instead often repaired erroneously by Alt-NHEJ, leading to genomic instability which could drive oncogenic transformation and progression. However, at the same time, these events represent an Achilles’ heel of cancer which could be therapeutically exploited by a synthetic lethality approach and could also offer new biomarker of responsiveness to immunotherapy. DSB, double strand break; DDR, dNA damage response; HR, homologous recombination; C-NHEJ, classical-non homologous end joining; Alt-NHEJ, alternative-non homologous end joining.

**Table 1 cancers-13-01392-t001:** PARPis approved for the treatment of progressive disease or maintenance after chemotherapy in HRD cancers (pivotal trials).

Drug	Ovarian Cancer	Breast Cancer	Pancreatic Cancer	Castration-Resistant Prostate Cancer
Olaparib	Treatment:-Germline BRCA 1/2 mutations-Following 3 or more line of therapy(NCT01078662)Maintenance:-Germline or somatic BRCA 1/2 mutations (first-line maintenance)-Also recurrent cancer without BRCA mutations-Advanced cancer with complete or partial response to first-line platinum-based chemotherapy(NCT01874353)	Treatment:-Germline BRCA 1/2 mutations-HER2 negative metastatic patients treated with chemo-Hormone receptor positive cancer treated with endocrine therapy or inappropriate for endocrine therapy(NCT02000622)	Maintenance therapy:-Germline BRCA 1/2 mutations (first-line maintenance)-Metastatic pancreatic adenocarcinoma with no disease progression on at least 16 weeks first-line platinum-based chemotherapy (NCT02184195)	Treatment:-HRR gene mutations -Metastatic castration-resistant prostate cancer who had disease progression while receiving enzalutamide or abiraterone.(NCT02987543)
Rucaparib	Treatment:-Germline BRCA 1/2 mutations-Following 2 or more rounds of chemotherapy(NCT01891344)Maintenance:-Recurrent cancer with complete or partial response to platinum-based chemotherapy-Regardless of BRCA mutation(NCT01968213)			Treatment:-adult patients with a deleterious BRCA mutation (germline and/or somatic)-metastatic castration-resistant prostate cancer (mCRPC) who have been treated with androgen receptor-directed therapy and a taxane-based chemotherapy.(NCT02952534)
Niraparib	Treatment:-Homologous recombination deficiency positive status:BRCA mutation orGenomic instability and progression >6 months after response to the last platinum-based chemotherapy-Advanced cancer treated with 3 or more rounds of chemotherapy(NCT02354586)Maintenance:-Recurrent cancer with complete or partial response to platinum-based chemotherapyRegardless of BRCA mutation(NCT02655016)			
Talazoparib		Treatment:-Germline BRCA1/2 mutations-HER2 negative metastatic cancer patients(NCT01945775)		
Veliparib	Treatment and maintenance:-Veliparib With Carboplatin and Paclitaxel -Newly Diagnosed Stage III or IV, High-grade Serous, Epithelial Ovarian, Fallopian Tube, or Primary Peritoneal Cancer(NCT02470585)

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
