# Peer review of "Alternative Non-Homologous End-Joining: Error-Prone DNA Repair as Cancer’s Achilles’ Heel"

_cancers, 2021, doi:10.3390/cancers13061392_

Round 1
Reviewer 1 Report
This manuscript by Caracciolo et al. aims to offer an overview of alternative non-homologous end joining pathway (Alt-NHEJ) as a promoter of genomic instability. Alt-NHEJ serves as a backup pathway when DSB repair is compromised. This paper complements recent reviews (doi.org/10.1038/nrm.2017.48; doi.org/10.1007/978-3-319-75836-7_15, etc.) and adds some new value. The review is well written, but there are a number of points that I would like to see in the final version, as well as a number of small remarks requiring the attention of the authors.
- It is suggested for the authors to underline the fact that Alt-NHEJ is more error prone than canonical NHEJ due to the increased likelihood of inappropriate joining of any two partially resected DNA ends.
- It is also suggested to discuss Fanconi anemia (FA) pathway in relation to Alt -NHEJ. It seems like FA can balance canonical and Alt -NHEJ because depletion of FA-protein decreases Alt -NHEJ with concomitant increase of Ku-mediated canonical NHEJ. An epistatic relationship between the FA pathway and translesion synthesis is well documented too. PARPis were suggested for FA-derived cancers.
- Table 1: Although not mandatory, the authors may want to include Veliparib, which is currently in clinical trials. Unlike Olaparib, this PARPi does not arrest cells in S phase and therefore may have clinical benefits.
Minor issues:
Figure 1
Please correct the wording (forma, on; resec on; diges on; etc ).
Any reason for depicting Ligase 3 with different color?
P4. Para 2:
ALT-NHEJ or Alt-NHEJ ?
Should be “mass spectrometry”
MALAT1 or MALAT-1?
Please check spaces throughout the text
P4 Para 3
Should be “DNA damage”
P5: once abbreviated, e.g. homologous recombination (HR) or PARP inhibitors (PARPis), use the abbreviation(s) throughout the text
Figure 2 – please increase the font size as some words require microscope
Author Response
This manuscript by Caracciolo et al. aims to offer an overview of alternative non-homologous end-joining pathway (Alt-NHEJ) as a promoter of genomic instability. Alt-NHEJ serves as a backup pathway when DSB repair is compromised. This paper complements recent reviews (doi.org/10.1038/nrm.2017.48; doi.org/10.1007/978-3-319-75836-7_15, etc.) and adds some new value. The review is well written, but there are a number of points that I would like to see in the final version, as well as a number of small remarks requiring the attention of the authors.
- It is suggested for the authors to underline the fact that Alt-NHEJ is more error-prone than canonical NHEJ due to the increased likelihood of inappropriate joining of any two partially resected DNA ends.
It is also suggested to discuss the Fanconi anemia (FA) pathway in relation to Alt -NHEJ. It seems like FA can balance canonical and Alt-NHEJ because depletion of FA-protein decreases Alt-NHEJ with a concomitant increase of Ku-mediated canonical NHEJ. An epistatic relationship between the FA pathway and translesion synthesis is well documented too. PARPis were suggested for FA-derived cancers.
We thank the Reviewer for the careful evaluation of our work. As suggested, we have amended the text in order to better describe the error-prone mechanisms of Alt-NHEJ and its relation with Fanconi anemia (FA) pathway (Para 1.2, 2.1 and 3).
- Table 1: Although not mandatory, the authors may want to include Veliparib, which is currently in clinical trials. Unlike Olaparib, this PARPi does not arrest cells in S phase and therefore may have clinical benefits.
We completely agree with the Reviewer. Accordingly, we have now included Veliparib in a new version of the Table 1.
Minor issues:
-Figure 1: Please correct the wording (forma, on; resec on; diges on; etc ). Any reason for depicting Ligase 3 with different color?
-P4. Para 2: ALT-NHEJ or Alt-NHEJ ? Should be “mass spectrometry”; MALAT1 or MALAT-1?
-Please check spaces throughout the text
-P4 Para 3: Should be “DNA damage”
-P5: once abbreviated, e.g. homologous recombination (HR) or PARP inhibitors (PARPis), use the abbreviation(s) throughout the text
-Figure 2 – please increase the font size as some words require microscope
We thank the Reviewer for all these suggestions. Accordingly, we have amended the text to avoid confusion and improve Figures’ readability.
Reviewer 2 Report
This review manuscript entitled “Alternative Non-Homologous End-Joining: Error-Prone DNA Repair as Cancer Achilles’ Heel” by Caracciolo, et al summarized the major relationship between the Alt-NHEJ pathway and cancers, which provides an useful overview about the effects of changing Alt-NHEJ regulation on cancer genesis and treatment. In order to present the content more logically, this reviewer suggests the authors to reorganize the structure: Remove “2.3 Drugging Alt-NHEJ repair” from subtitle “2. Alt-NHEJ and De-Regulation in Cancer” and relocate it as “4.1” under a new subtitle “4. Targeting Alt-NHEJ Repair”. Change “4. Alt-NHEJ Could Stimulate Immune Recognition of Tumor Cells” to “4.2”
Minor issue: “c-NHEJ” in figure 2 legend should be “C-NHEJ”, as those shown in the main text.
Author Response
1.This review manuscript entitled “Alternative Non-Homologous End-Joining: Error-Prone DNA Repair as Cancer Achilles’ Heel” by Caracciolo, et al summarized the major relationship between the Alt-NHEJ pathway and cancers, which provides a useful overview about the effects of changing Alt-NHEJ regulation on cancer genesis and treatment. In order to present the content more logically, this reviewer suggests the authors to reorganize the structure: Remove “2.3 Drugging Alt-NHEJ repair” from subtitle “2. Alt-NHEJ and De-Regulation in Cancer” and relocate it as “4.1” under a new subtitle “4. Targeting Alt-NHEJ Repair”. Change “4. Alt-NHEJ Could Stimulate Immune Recognition of Tumor Cells” to “4.2”
We completely agree with the Reviewer. Accordingly, we have moved the section Drugging Alt-NHEJ repair, under the new subtitle Targeting Alt-NHEJ Repair. We also changed the number of paragraphs of “Alt-NHEJ Could Stimulate Immune Recognition of Tumor Cells” from 4 to 4.2.
- Minor issue:
“c-NHEJ” in figure 2 legend should be “C-NHEJ”, as those shown in the main text.
We thank the Reviewer for the careful evaluation of our manuscript. As suggested, we have amended the text to avoid confusion and improve Figure readability.